# Kinetics of Moisture Loss and Oil Absorption of Pork Rinds during Deep-Fat, Microwave-Assisted and Vacuum Frying

**DOI:** 10.3390/foods10123025

**Published:** 2021-12-06

**Authors:** Hong-Ting Victor Lin, Der-Sheng Chan, Yu-Hsiang Huang, Wen-Chieh Sung

**Affiliations:** 1Department of Food Science, National Taiwan Ocean University, Keelung 202301, Taiwan; HL358@mail.ntou.edu.tw (H.-T.V.L.); 10832008@mail.ntou.edu.tw (Y.-H.H.); 2Center of Excellence for the Oceans, National Taiwan Ocean University, Keelung 202301, Taiwan; 3Department of Information Technology, Lee-Ming Institute of Technology, New Taipei City 243083, Taiwan; dschan@ms58.hinet.net

**Keywords:** pork rinds, vacuum frying, microwave-assisted frying, deep-fat frying, kinetics of oil absorption

## Abstract

The fat content of fried pork rinds is high, and alternative frying helps reduce the oil content and maintain their texture and taste. Different frying methods such as microwave-assisted, traditional deep frying and vacuum frying on the breaking force, color, microstructure, water loss and oil absorption attributes of fried pork rinds were evaluated in this study. The fat content of microwave-assisted and vacuum-fried pork rinds was lower (24.2 g/100 g dry weight basis (db) and 17.1 g/100 g db, respectively) than that (35.6 g/100 g db) of traditional deep-fat frying. Non-uniform, holy and irregular surface microstructures were obtained by vacuum frying due to rapid mass transfer at low pressure. The first-order kinetic models of water loss and oil absorption of traditional and microwave-assisted frying of pork rinds were established. Microwave frying caused a faster moisture loss rate, shorter frying time and lower pork rind oil content, makes it an attractive substitute for traditional deep-fat frying.

## 1. Introduction

Frying is a traditional and straightforward cooking method with a complex process that causes foods to absorb oil/fat and lose water. Simultaneously, chemical reactions including hydrogenation, oxidation and polymerization may occur [1]. The high temperature of frying oil, frequently (180–200 °C) produces a unique aroma called reaction flavor, including alcohols, aldehydes, carboxylic acids, furans, hydrocarbons, ketones, pyrazines and pyridines, which provides fried food a distinct flavor and taste [2]. Potato chips and French fries are popular and investigated snacks and fried food. Additionally, fried pork skin (pork rind) is a widely consumed traditional side dish or snack in Thailand [3]. Pork skin is deep-fat fried in oil to create the desired huge volume expansion, softness and crispness. The consumption of fried pork rind was estimated to be about USD 100,000,000 in the north of Thailand [4]. Although consuming fried food is usually related to the risk of health effects, involving diabetes [5], cardiovascular disease [6], hypertension and obesity [7], the consumption of fried foods has increased since many centuries ago. In addition, oil is one of the most expensive components and the food industry often recirculates the frying oil and tries to extend its life cycle and to reduce the oil absorption into food. Due to the disadvantages, consumers, food scientists and industries eagerly purchase, invent and generate low-fat snacks that maintain fried products’ appropriate texture and flavor.

A microwave fryer was developed to combine convective and microwave heating to higher turbulence and to reduce the frying time and oil content of food with thick dimensions. This frying equipment also generates microwaves to enhance moisture evaporation due to pressure-driven flow and reduced oil absorption due to less frying time than conventional deep-fat frying [8]. Vacuum frying is used in the food industry to decrease oil absorption and increase desired flavor and texture characteristics. Vacuum frying works at a reduced temperature (100–144 °C) and pressure (below 6.65 kPa) to minimize browning reaction and oil content during processing [9]. However, these new frying techniques have not been applied to pork rinds yet. Therefore, there are few scientific reports on fried pork rinds using microwave-assisted and vacuum frying to date [10].

The aim of this study was to investigate the effect of microwave-assisted and vacuum frying on oil absorption and the physical changes in fried pork rinds, as compared with traditional deep-fat frying. To our knowledge, there has been a limited number of studies on the mass transfer and texture characteristics of fried pork rinds during frying. Therefore, pork rinds were studied to understand mass and heat transfer kinetics in the process of frying for developing low-fat pork rind snacks.

## 2. Materials and Methods

### 2.1. Dried Pork Skin Preparation

Frozen skin of pork (*Sus scrofa domestica*) was obtained from Tai An Food Enterprise Co., Ltd. (Kaohsiung City, Taiwan). Frozen pork skin samples used for the experiments were of the similar characteristics and were thawed under running water and then cooked in boiling water for 1 h. Inner skin fat was scrapped, and the skin was cut into 5 cm × 2 cm strips and dried at 50 °C for 12 h, and then dried pork skin pieces were sealed in polyethene (PE) bags and maintained at room temperature until frying within 3 d.

### 2.2. Traditional, Microwave-Assisted and Vacuum Frying

Dried pork skin (30 pieces around 60 g each) was deep-fried in refined palm olein oil in an electric fryer at 180 °C [11]. Pork skin samples were fried at 0–5 min, and the samples were withdrawn from the frying process at the time intervals, 0.5, 1, 1.5, 2, 3, 4 and 5 min. At each time interval, at least three pork rind samples were taken out, and each was considered individually. Microwave-assisted frying was conducted using a 16 L microwave fryer (MF-1K, Chin Ying Fa Mechanicalind Co., Ltd., Chang Hua Hsien, Taiwan) at an intensity setting of P3 [12]. The microwave frying time and temperature were the same as the deep-fat frying. The pork rinds were immediately centrifuged to remove extra oil at 236 rpm for 1 min after traditional and microwave-assisted frying under atmospheric conditions.

A vacuum fryer was employed for vacuum frying; first, dried pork skin (20 g about ten strips) was vacuum-fried for 4, 8, 12, 16, 20 or 24 min (refined palm olein oil at 120 °C, 0.058 Mpa). Then, the samples were centrifuged at 300 rpm for 1 min under the same vacuum conditions (0.058 Mpa) to remove extra oil [13].

### 2.3. Lipid, Moisture Content and Water Activity

The lipid content of the fried pork rind samples was conducted following the diethyl ether extraction method using a Tecator Soxtec System HT1043 (Foss Analytical Co., Ltd., Hillerod, Denmark) at 96 °C for 4 h [14]. The moisture content of fried pork rinds was conducted following AOAC method 984.25 by using a convection oven drying at 105 °C. The wet weight basis in this study was defined based on pork rind not being dried to remove water. Dry weight basis (db) is reported based on pork rind samples dried at 105 °C to a constant weight. The water activity of fried pork rinds was measured following the method described by Mathlouthi [15]. A fried pork rind piece of around 2 g was put in the sample chamber at 25 °C for each measurement.

### 2.4. Mathematical Modelling

A uniform water distribution within each dried pork skin and constant oil temperature during frying was assumed in the samples, and the oil and water contents were considered independent. Krokida et al. [16] proposed the oil uptake of potato strips during frying as a first-order kinetic model Y = Y_e_ (1 − exp (−K_y_t)), and it was modified to the following equation due to pork skin containing oil when t = 0.
Y = (Y_e_ − Y_0_)(1 − exp(−K_y_t)) + Y_0_
(1)
where: t = frying time (min); Y = oil content at the frying time t (kg/kg db); Y_0_ = oil content at t = 0 (kg/kg db); Y_e_ = oil content at infinite frying time (kg/kg db); K_y_ = rate constant of oil absorption (min^−1^).

The moisture loss of potato strips during frying was described as a first-order kinetic model and the mass transfer phenomena was calculated using the formula below [16].
(2)$$\frac{d\left(X\right)}{\mathrm{dt}}=-K_{x}\;\left(X\;-{\;X}_{e}\right),\;\frac{\left(X\;-X_{e}\right)}{\left(X_{0}-X_{e}\right)}=\;\exp\;\left(-{\;K}_{x}t\right)$$
where: t = frying time (min); X = the moisture content at time t (kg/kg db); X_0_ = the moisture content at t = 0 (kg/kg db); X_e_ = the moisture content at infinite time (kg/kg db); K_x_ = the rate constant of moisture loss (min^−1^).

### 2.5. Breaking Force of Fried Pork Rinds

The breaking force of fried pork rinds was tested following the method reported by Su et al. [12]. Texture Analyzer RapidTA^®^+ (Horn Instruments Co., Ltd., Taichung, Taiwan) equipped with a spherical probe (P/0.5 s) (pre-test speed: 2.0 mm/s; test speed: 5.0 mm/s; post-test speed: 10 mm/s). The fried pork rind was placed on the testing platform of the Texture Analyzer. The trigger force was set at 5 g, with a deformation ratio of 100%. The maximum force (N) was determined as the breaking force measured during compression. The experiment was carried out three times, and an average value is reported.

### 2.6. Puffing Ratio of Fried Pork Rinds

The puffing ratio of fried pork rinds was determined as described in Chen et al. [17] and calculated as follows:(V_2_/W_2_) × 100%/ (V_1_/W_1_)(3)
where W_2_ = the weight of fried pork rinds; V_2_ = the volume of fried pork rinds; W_1_ = the weight of dried pork skin; V_1_ = the volume of dried pork skin.

### 2.7. Color

The International Commission on Illumination (CIE) L*a*b* color of fried pork rinds was recorded using a spectrocolorimeter (TC-1800MK-Ⅱ, Denshoku, Tokyo, Japan), as described in Le et al. [18]. The L* value represents the lightness of the color and it ranges from 0 to 100, L* = 0 signifies black and L* = 100 signifies white. Values a* and b* range from −60 to 60, with positive and negative a* representing green and red, respectively, and positive and negative b* representing yellow and blue, respectively. A white reference plate (X = 79.2, Y = 80.7, Z = 90.7) and a black cup were used as a standard before each round of tests. Color change (ΔE) was determined as following the formula:ΔE = [(△L*)^2^+(△a*)^2^+(△b*)^2^]^1/2^
(4)
where: △L* = L*_sample_ − L*_dried pork skin_; △a* = a*_sample_ − a*_dried pork skin_; △b* = b*_sample_ − b*_dried pork skin_. Each set of data was obtained and tested in triplicate.

### 2.8. Scanning Electron Microscopy (SEM)

Pork rinds of the optimal frying time for traditional deep-fat frying (5 min), microwave-assisted frying (3 min) and vacuum frying (20 min) were cut using a razor blade after lipid content testing. The samples were freeze-dried and adhered onto brass stubs and sputter-coated with platinum for 30 s (Hitachi E-1010, Tokyo, Japan). The sample surface was examined under a scanning electron microscope (Hitachi FE-SEM S3400, Tokyo, Japan) at 15 kV at 50 and 3000× magnifications, and representative images were chosen and photographed [19].

### 2.9. Statistical Analysis

Data were analyzed by one-way analysis of variance (ANOVA) and tested the general linear model (GLM) using the statistical package for social science (SPSS v.23.0) for Windows (SPSS Inc., Chicago, IL, USA). Ducan’s multiple range test was carried out to detect differences in means at a 5% significance level (*p* < 0.05). Pearson correlation analysis was used for linear correlations (*p* < 0.01). The mathematical modelling of oil and moisture content kinetics was determined by using Statgraphics 18 by the residual sum of squares (RSS).

## 3. Results

### 3.1. Oil Absorption, Moisture Content, Water Activity of Fried Pork Rinds and Kinetic Model of Oil Uptake and Water Loss

The oil content of fried pork rinds by different frying methods at various times is shown in Figure 1A. The moisture in the pork skin was evaporated and replaced by oil after 3 min deep-fat and microwave assisted frying (Figure 1A). In vacuum frying at 120 °C, oil is removed under vacuum-centrifuged conditions [9], and it reaches 15.17 g/100 g db after vacuum frying for 4 min and 17.43 g/100 g db for 24 min (Figure 1A). Vacuum-fried pork skin results in an oil content of 15.17 g/100 g db after 4 min frying at 120 °C under vacuum (Figure 1A). In addition, it was necessary to use de-oiling under vacuum to remove the absorbed oil on the surface of fried pork rinds before oil cooling [9]. The fat content of fried pork rinds varied between 35.63 g/100 g db for traditional deep-fat frying and 24.21 g/100 g db for microwave-assisted frying (Figure 1A). Vacuum frying exhibited a lower oil content of 17.14 g/100 g db, as compared with deep-fat and microwave-assisted frying, respectively. Traditional and microwave-assisted frying was operated at 180 °C, and de-oiling processing was conducted under atmospheric conditions. Microwave-assisted frying under microwave heating resulted in an oil content of 24.70 g/100 g db after 5 min (Figure 1A).

The water content of dried pork skin was 19.39% (% wet basis; wb), and it decreased rapidly during the first 5 min of deep-fat (3.08% wb), microwave-assisted (after 3 min to 3.57% wb) and vacuum frying (after 20 min to 5.36% wb), which is because of the direct contact of pork skin with high-temperature frying oil (Figure 1B). Vacuum frying demonstrated slow dehydration in the first 4 min and reached around 5.36% wb in 20 min. The water content reduced to 3.08% and 3.57% wb in 5 and 3 min, respectively, in the deep-fat and microwave-assisted frying. Moisture evaporating with frying time for three frying methods are shown in Figure 1B. Nevertheless, the water content of fried pork skin decreased fast in microwave-assisted frying following traditional deep-fat frying than vacuum frying in the first 4 min (Figure 1B).

The water activity of dried pork rinds by deep-fat and microwave-assisted frying decreased more rapidly than vacuum frying in the first 4 min due to the thermal transfer at 180 °C, and absorbed oil replacing the capillary interstice of pork skin was fast (Figure 2).

As for the first-order oil absorption phenomena with the pork skin frying process, Krokida et al. [16] assumed that fresh potato oil content is zero; nevertheless, the oil content of pork skin is 10.67% wb [18]. Therefore, the oil content of raw pork skin cannot be omitted. The proposed equation was modified as in Equation (1).

The oil absorption kinetic model of vacuum frying could not be predicted by Equation (1) because the oil content of fried pork rinds did not change significantly after 4 min of frying (Figure 1A). Figure 1C only has two predicted curves that appear and not three as in the other graphs of Figure 1. The oil absorption and predicted kinetic models (Equation (1)) of fried pork rind by deep-fat and microwave-assisted frying are shown in Figure 1C. The predicted respective coefficient of determination (R^2^) of deep-fat and microwave-assisted frying is higher than 0.95 (Table 1). It indicates that the expected first-order oil absorption kinetic model (Equation (1)) fits the data collected. The rate constant of oil uptake (K_y_), the oil content at equilibrium frying time (O_e_) and the oil content at zero time (O_o_) of traditional frying method were higher than those of microwave-assisted frying method (Table 1).

A first-order water loss kinetic model (Equation (2)) proposed by Krokida et al. [16] for frying potato strips was also used to show the water loss phenomena during the pork skin frying process. The results of dehydration curve estimation for the water loss of pork skin by three frying methods are shown in Figure 1D. Parameters estimation for three frying methods are listed in Table 2. The rate constant of moisture loss (K_x_, min^−1^) calculated from the microwave-assisted and deep-fat frying was higher (1.14 and 0.68 min^−1^, respectively) than that of the vacuum frying method (0.13 min^−1^) (Table 2). It indicated a faster mass transfer of water owing to higher heat mass transfer by higher frying temperature than vacuum frying oil.

### 3.2. Breaking Force, Puffing Ratio and Color of Frying of Fried Pork Skin

The water of dried pork skin was evaporated and replaced with hot oil during frying, contributing to the aroma and flavor (sensory evaluation data not shown) of fried pork rinds and the hardness and crispness of the fried pork rinds. Fan et al. [20] also found breaking force is associated with fried carrot chip hardness and crispness. Additionally, the breaking force of a carrot chip could decrease by increasing the vacuum frying oil temperature. Figure 3 shows the breaking force of fried pork skin using different frying methods and time. Breaking force decreased during frying, and the water content decreased to around 6%; the breaking force increased for further frying as water content reduced to about 3%.

The puffing ratio showed 401%, 408% and 226% increase for deep-fat, microwave-assisted and vacuum frying at frying times of 5 min, 5 min and 24 min, respectively, for the dried pork skin (Table 3). The puffing ratio of fried pork rinds increased to a maximum ratio by deep-fat and microwave assisted frying (*p* < 0.05; Table 3). The puffing ratio of frying time reported in minute 4 confirmed both deep-fat frying and microwave-assisted frying at 180 °C had better puffing ratio (379% and 406%, respectively) than that of vacuum frying at 120 °C and 0.058 Mpa (226%).

Color changes (lightness, redness and yellowness) in fried pork rinds were observed as the dried pork skin was fried (Table 4 and Appendix A). There were differences in color values of fried pork rinds and those in the dried pork skin. Higher L* and lower a* values for fried pork rinds were found during frying compared with the dried pork skin (Table 4).

### 3.3. Characteristics and Microstructure of Fried Pork Rinds

The optimal frying time for traditional deep-fat, microwave-assisted and vacuum frying for pork rinds was 5, 3 and 20 min, respectively (Table 5). Comparisons of the physical characteristics of fried pork rinds by different frying methods under similar water content (3–5%) are shown in Table 5. Various frying methods had significant effects on the oil content of fried pork rinds. The microwave-assisted frying decreased the frying time and oil content of pork rinds as advertised. Nevertheless, traditional deep-fat and microwave-assisted fried pork rinds contained higher oil content (35.6% and 24.2%, respectively) than the vacuum-fried (17.1%).

Non-uniform and irregular surface structures of fried pork skins were observed on the traditional deep frying and microwave-assisted frying (Figure 4A,C) compared with the surface of vacuum-fried pork skin (Figure 4E).

In addition, the drier exterior and outer surface mitigated the water removal rate in vacuum-fried pork rinds at 120 °C. Gradual water and heat transfer prevented the disruption of the cellular matrix and sustained the structure of the cell, particularly for the outer surface (Figure 4).

### 3.4. Correlations between Physical Properties of Fried Pork Rinds

Correlation analysis was performed between fried pork rinds’ physical properties to better comprehend the relationship among different quality characteristics (Appendix A). All frying methods showed a negative relation among breaking force, moisture content, water activity and frying time of fried pork skin; however, in the tested frying methods, the puffing ratio, CIE L* and △E were positively correlated with the frying time. There is a high positive correlation among appearance, flavor, texture and overall acceptability. Additionally, there is a high positive correlation between unpleasant smell and greasy sensation of fried pork rinds (Appendix A; *p* < 0.01).

## 4. Discussion

### 4.1. Oil Absorption, Moisture Content, Water Activity of Fried Pork Rinds and Kinetic Model of Oil Uptake and Water Loss

Microwave-assisted frying showed a faster dehydration rate than traditional deep-fat frying. The more rapid heat conduction of the microwave radiant energy for a microwave fryer exhibited faster heat transfer than traditional deep-fat frying. Microwaves will penetrate to a depth of more than 10 mm and are absorbed by the moisture content of pork skin [21]. Microwaves can generate heat within the pork skin, fat and collagen and water with different loss factors; therefore, different components do not immediately heat up equally. Nevertheless, as the component is heated up, it can conduct between pork skin components [21].

Low-temperature vacuum frying showed a slight decrease in the water activity and heated 24 min to get to 0.3 from a food safety perspective [22]. Microwave-assisted frying showed the fastest reduction in water activity in the first 4 min of frying, and a succeeding reduction was observed, similar to traditional deep-fat frying (180 °C). The faster initial decrease was because microwave radiant energy takes a shorter time to penetrate the pork skin than other heat-up methods.

The moisture in the dried pork skin played an important role as it evaporated during the frying process. The evaporated of steam water and hot oil cooked the denatured pork skin as it escaped through the pores because of internal pressure. When the fried pork skin was taken out of the hot oil, the temperature dropped, and reduced steam production inside pork skin and the fried pork skin temperature started to cool down. As the internal pressure decreased, the voids created by water evaporation through the pores absorbed the frying oil into the outer layer.

The oil absorption kinetic model (Equation (1)) of vacuum frying could not be predicted because the oil content of fried pork rinds did not change significantly after 4 min of frying (Figure 1A). The oil absorption curve of pork rinds demonstrated a different pattern in traditional deep-fat and microwave-assisted frying (Figure 1C), indicating that the microwave radiant energy in a microwave fryer could lower oil absorption and shorten frying time as it was proposed—it was confirmed by the parameters, the rate constant of oil uptake (K_y_), the oil content at equilibrium frying time (O_e_) and the oil content at zero time (O_o_) of Table 1. The K_y_ and O_e_ of the traditional frying method were higher than those of the microwave-assisted frying method, indicating higher oil uptake rate and oil content at the same frying time (Table 1). A similar pattern of oil uptake was also proposed in potato chips [12], which agreed with our results.

The water content at limitless frying time should be close to zero, and deep-fat and microwave-assisted frying had an ideal value of 0.03 (g/g db; X_e_); in addition, vacuum frying processed at 120 °C exhibited a higher water content (0.05 g/g db; X_e_). The kinetic model of water loss (Equation (2)) was illustrated in Figure 1D. The estimated water content by the proposed model was demonstrated as successive lines. A fast reduction in the water content for all pork skins was observed during frying. The equilibrium water content of fried pork rinds was in a range of 0.23 to 0.25 kg/kg db (X_0_), lower than the results of potato strips (0.6–0.8 kg/kg db) reported by Krokida et al. [16]. These data showed that higher equilibrium water content was obtained for starch-based potato strips than protein-based pork rinds.

### 4.2. Breaking Force, Puffing Ratio and Color of Frying of Fried Pork Skin

The breaking force of fried pork skin decreased during frying before the water content decreased to 6% because of the dehydration of collagen and protein. Furthermore, the breaking force of fried pork skin raised by increased frying time when the water content reached 3% (Figure 1B and Figure 3), denaturing other substances and proteins in the pork skin. It can also be observed in Figure 3 that the breaking force could increase with extended frying time (from 2.5 to 5 min) in both microwave-assisted and traditional deep-fat frying. Pork skin breaking force in the first 2 min of frying was high because its moisture content is high (>6%), and it has a low puffing ratio. A similar texture change trend was found in vacuum frying at a longer frying time pattern (Figure 3), also reported for fried fish skins [13]. Traditional and vacuum frying offered less hard fried pork rinds with breaking force about 49–54 N (at a frying time of 2.5 and 20 min, respectively), which was firm for a snack texture. This was because of the prolonged frying time at 120 °C, hardening the surface of pork skin in vacuum frying. The breaking force increased to about 80 N at 24 min. Therefore, it is not recommended to extend the frying time for vacuum and traditional deep frying after optimal frying. A firmer fried pork skin (over denaturing collagen and protein) would be obtained by extending the frying time or by frying at a higher temperature in traditional deep frying or vacuum frying.

Water evaporation vigorously occurs on the surface of pork skin after dipping into the higher temperature frying oil. The evaporation of water moves towards the interior of the dried pork skin, and dehydrated and denatured pork skin crust is formed. Additionally, the water evaporation slows down and reaches a 3% moisture content bubble-end point. The puffing structure was set by protein denaturing; vacuum frying is another case, and water evaporation occurs on the surface of the dried pork skin. As vacuum frying progresses, denatured, dehydrated pork skin crust is formed when the chamber temperature is over the boiling temperature inside the vacuum fryer. Additionally, water starts evaporating vigorously to puff the structure of dried pork skin until the dehydrated and denatured protein reaches a moisture content of around 3% (Table 3).

It may be the puffing effect resulting from oil absorption, heat-triggered browning and Maillard reactions that lighten pork skins. Although the vacuum-frying process at a lower temperature (120 °C) and less oxygen reacts with protein, it did not significantly influence the L* value of fried pork rinds at optimal frying time (5 min for traditional deep frying; 3 min for microwave-assisted frying and 20 min for vacuum frying) (Table 4). The results of fried fish skin reported by Fang et al. [13] were different for vacuum frying, which may be because at 120 °C, the heat transfer of oil and temperature on the surface were lowered. The color difference (△E) values of vacuum-fried pork skin compared with the value of the dried pork skin at 4 min frying were lower than those of traditional and microwave-assisted frying methods (Table 4; *p* < 0.05). Both L* values (lightness) of 4 min fried pork rinds by traditional deep frying and microwave-assisted frying were higher than that of 4 min vacuum-fried pork skin (Table 4). This may be due to the heat transfer of lower oil temperature, which is not high compared with other frying methods at the same frying time. The total color difference slowly increased after 24 min of vacuum frying (Table 4).

The microwave-assisted frying method demonstrated that it took less time to decrease the moisture content of fried pork skin to 3% and to obtain a lighter color appearance (Table 5). Although it was not the lowest oil content of fried pork, it significantly reduced oil content of fried pork skin, frying time and energy and produced a high puffing ratio product.

### 4.3. Characteristics and Microstructure of Fried Pork Rinds

During deep-fat frying, hot oil transferred energy to water molecules and activated them. Microwave-assisted frying enhanced energy transfer via the cell protein matrix, partially causing the structure disruption and forming irregular holes and cracks (Figure 4). The heat stream from the vacuum frying was less to transfer into the dried or partially dried surface of pork skin at a lower frying temperature (120 °C).

However, we found more cracks and holes were on the surface of traditional deep-fried pork skin (Figure 4B) compared with that of microwave-assisted fried pork skin (Figure 4D). Microwave-assisted frying took less time to fry, and fewer cracks and holes on the surface of fried pork skin might explain why it significantly absorbed less oil during frying. The SEM of vacuum-fried pork skin demonstrated more regular and uniform hole structures (Figure 4E,F). We believe the absorption of frying oil, collagen gelatinization, dehydration, protein denaturation and spontaneous evaporation of water during low temperature vacuum frying at 120 °C and 0.058 Mpa created this structure. The frying temperature could even go higher under vacuum to shorten frying time.

### 4.4. Correlations between Physical Properties of Fried Pork Rinds

There was a strong relationship between overall acceptability and fried pork rinds’ appearance, flavor and texture (Appendix A). Therefore, the desirable quality characteristics caused by different frying methods can be incorporated into fried pork rinds. Acceptable qualities of fried pork skin were related to different frying methods, which demonstrated the effect of studies on the frying methods for processing fried pork rinds with low fat and favorable texture. It indicated that the desirable quality properties of appearance, flavor and texture were associated with consumer preference. Therefore, this understanding can help to produce a fried pork skin with crispy texture and light appearance but low fat content.

## 5. Conclusions

The microwave-assisted frying method significantly decreased breaking force, optimal frying time (3 min) and oil content (24.2 g/100 g db) of fried pork rinds compared with traditional frying (5 min frying time and 35.6 g/100 g db oil content). The vacuum frying method decreased the puffing ratio and oil content (17.1 g/100 g db) of fried pork rinds but dramatically increased optimal frying time (20 min) and made the product hard; thus, frying at a lower frying temperature (120 °C) resulted in pork rinds that were unsuitable for consumption. The properties of pork rinds can be improved by frying at a higher temperature. This study indicated significantly different rate constants for oil absorption and water loss kinetic models of fried pork rind between microwave-assisted and deep-fat frying. Furthermore, it showed that microwave-assisted frying reduced frying time and decreased the fat content of fried pork rinds. Additionally, their respective coefficients of determination (R^2^) were more than 0.95; the influence of frying methods were demonstrated to be remarkable on the textural properties of fried pork rinds. The kinetic models confirmed were first-order, and they could be used for predicting the moisture and oil content changes during deep-fat and microwave assisted frying in the fried pork rind industry. Statistical correlation data showed that frying time, oil content, water activity, moisture content, breaking force, color and puffing ratio were highly related. Microwave-assisted frying significantly decreased oil content and frying time of pork rinds to develop healthier foods with less energy spent while obtaining the best appearance and texture, thus demonstrating it to be the most recommended technology for the fried pork rind industry.

## Figures and Tables

**Figure 1 foods-10-03025-f001:**
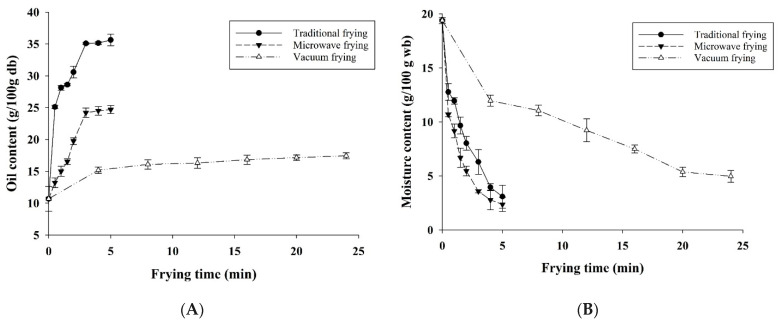
Oil uptake and moisture loss properties of pork skin during frying: (**A**) oil content; (**B**) moisture content; (**C**) kinetic model of oil uptake (Equation (1)); (**D**) kinetic model of moisture loss (Equation (2)).

**Figure 2 foods-10-03025-f002:**
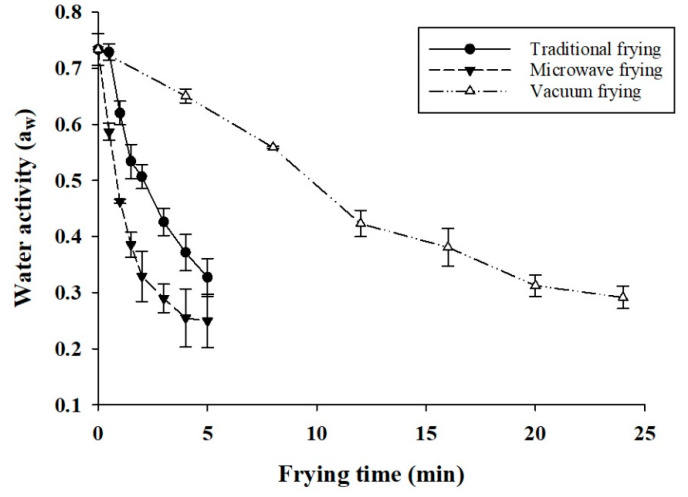
Comparison of the water activity (aw) of fried pork rind by different frying methods and frying time.

**Figure 3 foods-10-03025-f003:**
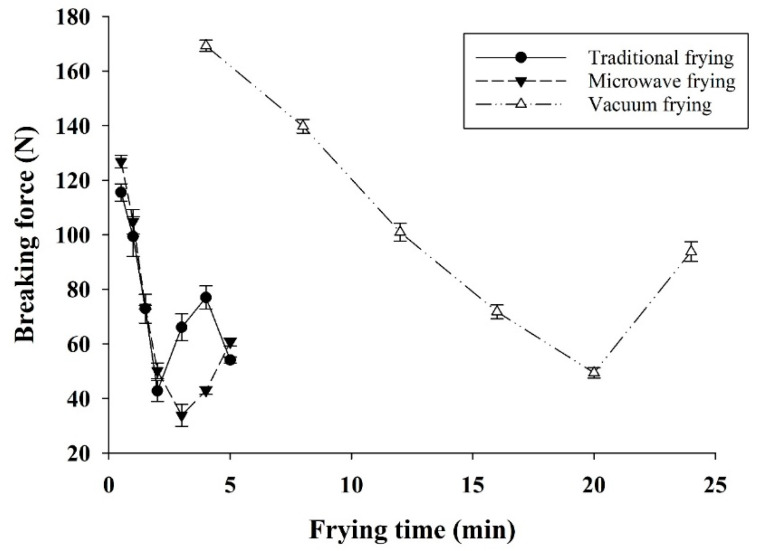
Observed breaking force of fried pork skins by various frying methods.

**Figure 4 foods-10-03025-f004:**
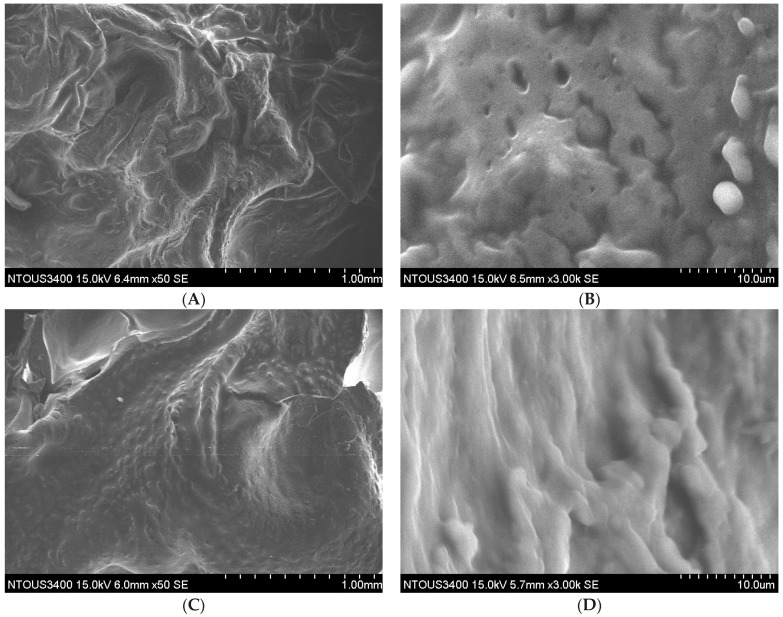
Scanning electron micrographs at 1.0 mm and 10.0 μm scale of fried pork skins of the optimal frying time for deep-fat frying (5 min), microwave-assisted frying (3 min) and vacuum frying (20 min): (**A**) and (**B**) deep frying; (**C**) and (**D**) microwave assisted frying; (**E**) and (**F**) vacuum frying. The magnification was set as 50×, 3000×.

**Table 1 foods-10-03025-t001:** Parameters of oil absorption model of fried pork rind for model Equation (1) with different frying methods (Krokida et al. 2006).

	K_y_	O_e_	O_0_	R^2^
Traditional deep frying	1.13 ± 0.16 ^b^	0.35 ± 0.01 ^b^	0.13 ± 0.01 ^b^	0.9537
Microwave-assisted frying	0.42 ± 0.07 ^a^	0.28 ± 0.01 ^a^	0.09 ± 0.00 ^a^	0.9715

Expressed as mean ± standard deviation (n = 3). Values followed by different superscript letters are significantly different (*p* < 0.05) between different frying methods. K_y_: the rate constant of oil uptake (min^−1^). O_e_: the oil content at equilibrium frying time (g/g db). O_0_: the oil content at zero time (g/g db). R^2^: coefficient of determination.

**Table 2 foods-10-03025-t002:** Parameters of water loss model of fried pork rind for model Equation (2) with different frying methods (Krokida et al. 2006).

	K_x_	X_e_	X_0_	R^2^
Traditional deep frying	0.68 ± 0.27 ^b^	0.03 ± 0.02	0.23 ± 0.03	0.9613
Microwave-assisted frying	1.14 ± 0.11 ^c^	0.03 ± 0.00	0.23 ± 0.00	0.9665
Vacuum frying	0.13 ± 0.00 ^a^	0.05 ± 0.00	0.23 ± 0.02	0.9557

Expressed as mean ± standard deviation (n = 3). Values followed by different superscript letters are significantly different (*p* < 0.05) among different frying methods. K_x_: rate constant of moisture loss (min^−1^). X_e_: moisture content at equilibrium frying time (g/g db). X_0_: moisture content at zero time (g/g db). R^2^: coefficient of determination.

**Table 3 foods-10-03025-t003:** Puffing ratio of fried pork rind by different frying methods and time.

Puffing Ratio (%)							
Frying method	0.5 min	1 min	1.5 min	2 min	3 min	4 min	5 min
Traditional deep frying	216.3 ± 41.1 ^cd^	263.4 ± 18.0 ^e^	319.9 ± 9.6 ^f^	342.2 ± 14.1 ^fgh^	364.2 ± 12.8 ^ghi^	378.8 ± 12.0 ^hij^	400.8 ± 8.7 ^ij^
Microwave-assisted frying	206.8 ± 3.2 ^de^	304.9 ± 16.6 ^fg^	339.6 ± 13.0 ^fgh^	365.6 ± 5.2 ^ghij^	400.2 ± 6.4 ^ij^	405.6 ± 1.7 ^ij^	407.8 ± 2.0 ^j^
Vacuum frying	—	—	—	—	—	125.1 ± 6.1 ^a^	—
**Puffing Ratio (%)**							
	6 min	8 min	10 min	12 min	16 min	20 min	24 min
Traditional deep frying	—	—	—	—	—	—	—
Microwave-assisted frying	—	—	—	—	—	—	—
Vacuum frying	—	150.8 ± 10.4 ^ab^	—	177.7 ± 4.1 ^bc^	198.7 ± 2.6 ^cd^	224.0 ± 5.9 ^de^	226.3 ± 8.9 ^de^

Expressed as mean ± standard deviation (n = 3). Values followed by different letters are significantly different (*p* < 0.05).

**Table 4 foods-10-03025-t004:** Color differences of fried pork rind by different frying methods and time.

Sample	Frying Time	L *	a *	b *	△E
Dried pork skin	—	46.79 ± 2.87 ^a^	−0.85 ± 1.10 ^h^	36.02 ± 2.27 ^a^	—
Traditional deep	0.5 min	60.33 ± 2.51 ^bc^	−4.38 ± 0.33 ^efg^	39.09 ± 1.49 ^abc^	16.68 ± 3.32 ^b^
frying	1 min	66.92 ± 5.35 ^defgh^	−6.45 ± 1.31 ^abc^	37.00 ± 1.66 ^a^	22.14 ± 5.31 ^cdef^
	1.5 min	67.09 ± 4.16 ^defgh^	−6.50 ± 1.52 ^abc^	37.54 ± 1.84 ^ab^	22.40 ± 4.06 ^cdef^
	2 min	69.73 ± 2.89 ^fghij^	−6.56 ± 1.05 ^abc^	35.88 ± 4.56 ^a^	25.13 ± 2.67 ^efgh^
	3 min	70.60 ± 0.93 ^ghij^	−5.95 ± 0.66 ^bcde^	41.79 ± 2.62 ^bcdef^	26.54 ± 0.33 ^fghi^
	4 min	71.61 ± 0.68 ^hijk^	−7.65 ± 0.29 ^a^	38.76 ± 2.21 ^abc^	28.05 ± 0.99 ^ghij^
	5 min	71.22 ± 1.66 ^hijk^	−6.93 ± 1.57 ^ab^	39.15 ± 4.74 ^abc^	27.19 ± 1.27 ^fghij^
Microwave-assisted	0.5 min	64.66 ± 2.25 ^de^	−4.67 ± 0.81 ^defg^	45.06 ± 2.48 ^efg^	20.47 ± 2.69 ^bcde^
frying	1 min	68.03 ± 2.04 ^defghi^	−5.21 ± 0.41 ^cdef^	43.05 ± 2.07 ^cdefg^	22.88 ± 1.51 ^def^
	1.5 min	68.87 ± 1.67 ^efghi^	−4.64 ± 0.85 ^efg^	44.89 ± 0.70 ^efg^	24.11 ± 1.72 ^defg^
	2 min	72.01 ± 0.99 ^ijkl^	−5.24 ± 0.38 ^cdef^	44.42 ± 0.94 ^def^	26.96 ± 0.69 ^fghij^
	3 min	73.83 ± 1.83 ^jkl^	−4.71 ± 1.70 ^defg^	46.41 ± 2.09 ^g^	29.32 ± 1.51 ^hij^
	4 min	75.17 ± 1.02 ^kl^	−5.40 ± 0.60 ^bcdef^	47.09 ± 1.89 ^g^	30.85 ± 0.88 ^ij^
	5 min	76.15 ± 0.17 ^l^	−6.28 ± 0.30 ^abcd^	46.15 ± 0.68 ^fg^	31.53 ± 0.38 ^j^
Vacuum frying	4 min	57.94 ± 3.39 ^b^	−3.28 ± 0.43 ^g^	37.71 ± 2.74 ^ab^	11.71 ± 3.60 ^a^
	8 min	63.63 ± 5.95 ^cd^	−3.11 ± 0.67 ^g^	40.24 ± 2.64 ^abcd^	17.57 ± 4.45 ^bc^
	12 min	66.46 ± 2.38 ^defg^	−3.79 ± 0.39 ^fg^	41.47 ± 1.58 ^bcde^	20.65 ± 2.63 ^bcde^
	16 min	68.35 ± 1.38 ^efghi^	−3.97 ± 0.59 ^fg^	44.58 ± 1.07 ^defg^	23.43 ± 1.34 ^defg^
	20 min	70.28 ± 1.84 ^ghij^	−4.54 ± 0.92 ^efg^	44.73 ± 1.82 ^efg^	25.37 ± 1.94 ^efgh^
	24 min	70.45 ± 1.54 ^ghij^	−4.62 ± 1.12 ^efg^	44.30 ± 1.96 ^defg^	25.41 ± 1.79 ^efgh^

Expressed as mean ± standard deviation (n = 3). Values followed by different letters within each column are significantly different (*p* < 0.05). L* value represents the lightness of the color, L* = 0 signifies black and L* = 100 signifies white; a* representing green and red, with positive and negative a* representing green and red, respectively; and positive and negative b* representing yellow and blue, respectively; ΔE represents color difference.

**Table 5 foods-10-03025-t005:** Comparison of the frying time, moisture content, water activity, oil content, puffing ratio, color differences and puffing thickness of different frying methods on the basis of similar breaking force.

Frying Method	Frying Time	Breaking Force (N)	Moisture Content (g/100 g wb)	Water Activity	Oil Content (g/100 g db)	Puffing Ratio (%)
Traditional deep frying	5 min	54.09 ± 1.08 ^c^	3.08 ± 1.06 ^b^	0.33 ± 0.03 ^b^	35.63 ± 0.90 ^d^	400.8 ± 8.7 ^b^
Microwave-assisted frying	3 min	33.82 ± 4.04 ^a^	3.57 ± 0.14 ^b^	0.29 ± 0.03 ^b^	24.21 ± 0.73 ^c^	400.2 ± 6.4 ^b^
Vacuum frying	20 min	49.35 ± 1.86 ^c^	5.36 ± 0.44 ^c^	0.31 ± 0.02 ^b^	17.14 ± 0.44 ^b^	224.0 ± 5.9 ^a^
		L*	a*	b*	△E	Puffing thickness (cm)
Traditional deep frying	5 min	71.22 ± 1.66 ^ab^	−6.93 ± 1.57 ^a^	39.15 ± 4.74 ^a^	27.19 ± 1.27 ^ab^	0.69 ± 0.04 ^b^
Microwave-assisted frying	3 min	73.83 ± 1.83 ^b^	−4.71 ± 1.70 ^ab^	46.41 ± 2.09 ^b^	29.32 ± 1.51 ^b^	0.55 ± 0.08 ^a^
Vacuum frying	20 min	70.28 ± 1.84 ^ab^	−4.54 ± 0.92 ^ab^	44.73 ± 1.82 ^ab^	25.37 ± 1.94 ^ab^	0.58 ± 0.06 ^a^

Expressed as mean ± standard deviation (n = 3). Values followed by different letters within each column are significantly different (*p* < 0.05).

## Data Availability

The data presented in this work are available on request from the corresponding author.

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
