# Peer review of "Kinetics of Moisture Loss and Oil Absorption of Pork Rinds during Deep-Fat, Microwave-Assisted and Vacuum Frying"

_foods, 2021, doi:10.3390/foods10123025_

Round 1

Reviewer 1 Report

The term microwave frying is not clear in the title, suddenly using the term microwave assisted frying could be more appropriate?
In the first reference to the authors' email [email protected] (P.H.H) is mentioned, avoid the use of personal emails, the use of corporate / institutional emails is preferable.
In the summary there are parts that are not clear, it is necessary to revise the wording.
In the summary, it is necessary to define well what is meant by cooking time?
In the introduction, what happens with oil in frying processes could be further developed, for example it is not mentioned that in the process it is the most expensive component and that it is often recirculated ...
It is necessary to review the term reaction flavor, as it is written it is not clear
When talking about the result of the frying process, the development of the characteristic texture of this type of product can also be added.
In the sentence that speaks of the increase in fried foods, it is necessary to specify a time horizon within which this increase occurred.
When talking about the microwave frying process, it is necessary to develop this new concept in more depth, as it is written there is no specification for the development of the process or the characteristics of the devices used for this purpose. The article would improve the interest of the readers if it improves this point.
They could specify the ranges of temperatures and pressures used in vacuum frying.
When it is mentioned that there are few works regarding new technologies for frying pork rinds, it could include a reference to support that phrase.
In point 2.2
It is necessary to review the wording, the samples were fried at times xx, xx, and the samples were withdrawn from the frying process at the time intervals already mentioned?
It is not clear if the oil draining process was the same for all types of frying experiments?
Regarding the microwave frying process, there is nothing clear about the device used, if this is an experimental setup or a commercial device, layout, quantity of oil used, etc? In addition, it is mentioned that the power 3 of the equipment was used, which is not precise, it would be convenient for them to be expressed as a percentage of power in relation to 100% of power or also what power they actually used.
It is not clear whether the pigskin pieces used for the experiments were of the same characteristics.
In the moisture determination method 984.25 is mentioned but the year/version of the AOAC used is not established
In the breaking force tests, it is not explained in sufficient detail how these measurements were developed.
In all equations, the number corresponding to each one is shown twice.
It is necessary to specify in the color measurements which was the illuminant and the observer used.
The description of the procedures for microscopic observations is incomplete
It is necessary to review the scales especially of time, for example 1 A and B are the same but D is presented differently for the same time interval
It is not explained anywhere why at 1 C there are no measurements for vacuum frying
Check if the equations used for the kinetic models have any proper name and if there are more equations of similar phenomena such as dehydration that can be applied.
In figure 2 the concept of water activity must be written with a lowercase letter and use the subscript w.
In table 1 there is no information for vacuum frying? Why the document must be uniform ...?
In 3.2 they mention the contribution of frying to aroma and flavor, but didn't they really measure it?
What is the contribution of comparing with carrot chip, could you develop this point further and include other similar products in the discussion
Check the figure number, figure 1B does not refer to the breaking force
The discussion regarding the puffing ratio is very poor, the article would improve if it is broadened
Table 3 mentions traditional frying? Traditional deep-frying experiments were done, if so it should be described in the methods part.
It is important to review and consider that the color of this type of product is strongly influenced by the porosity, translucency, and brightness of the samples. The discussion of color needs to be improved.
What does table 4 refer to with cooked? Nowhere is there a talk of a cooking process?
To make the color analysis and its results more attractive, you could use a free tool like Adobe Color Wheel to better represent the evolution of the color of the samples.
Review the relevance of table 5, it is important not to repeat data.

The analysis of the microphotographs is very poor, it can be expanded and relate this to the texture / strength, oil content and moisture, in the same way the correlation analysis can be improved.
In some parts of the speech of steam cooked denaturated pork skin… which can be misleading, it is necessary to standardize all with pork rinds.

The conclusion can be improved by including the possibility of developing healthier foods, with less oil consumption and also the different use of energy and possible gaps to investigate such as energy aspects, stability and sensory evaluation of the products obtained.

Author Response

Responses to Comments and Suggestions for Authors

Foods

Title: Kinetics of moisture loss and oil absorption of pork rinds during deep-fat, microwave assisted and vacuum-frying

Dear Reviewer #1

Thank you for your instruction on revising abstract, materials and methods, results and discussion. conclusion. We have rewritten the manuscript by Dr. Lin accordingly, and replied to comments and suggestions for authors are listed below point:

Reviewer #1’s comments and suggestions:

Point 1: The term microwave frying is not clear in the title, suddenly using the term microwave assisted frying could be more appropriate?

Response 1: We have revised the title and texts of “microwave” to “microwave assisted” in the revised manuscript. Please see title highlighted in red and texts of the revised manuscript. Thanks for the suggestion.

Point 2: In the first reference to the authors' email [email protected] (P.H.H) is mentioned, avoid the use of personal emails, the use of corporate / institutional emails is preferable.
Response 2: We have replaced the school email address of Mr. Huang, [email protected], with his Gmail address. Please see the revised institutional email on page 1, line 6.

Point 3: In the summary there are parts that are not clear, it is necessary to revise the wording.

Response 3: We have rewritten and rechecked the abstract section in our article carefully as the red-marked texts in the revised manuscript. Thanks for the suggestions, and we very much appreciate your consideration on this matter. (Please see the abstract of the revised manuscript).

Point 4: In the summary, it is necessary to define well what is meant by cooking time?
Response 4: The phrase “shorter cooking time” was revised to “shorter frying time”. Thanks for the great comment. Please see the revised abstract on page 1, line 20.

Point 5: In the introduction, what happens with oil in frying processes could be further developed, for example it is not mentioned that in the process it is the most expensive component and that it is often recirculated ...

Response 5: The sentence of “In addition, oil is one of the most expensive components, and the food industry often recirculated the frying oil and tried to extend its life cycle and reduce the oil absorption into food” was added to the first paragraph of the introduction of the revised manuscript on page 1, lines 39-41. Thanks for the suggestion.

Point 6: It is necessary to review the term reaction flavor, as it is written it is not clear.

Response 6: The reaction flavor of deep-fried flavor compounds was added to the introduction and the cited paper has been replaced. Please see the introduction and reference section of the revised manuscript on page 1, lines 31, reference #2. Thanks for the great comment.

Point 7: When talking about the result of the frying process, the development of the characteristic texture of this type of product can also be added.

Response 7: Sentence “Pork skin is deep-fat fried in oil to create the desired huge volume expansion, softness, and crispness.” was added to explain the characteristic texture of fried pork skin. Please see the introduction of revised manuscript lines 34-35 on page 1. Thanks for the great suggestion.

Points 8: In the sentence that speaks of the increase in fried foods, it is necessary to specify a time horizon within which this increase occurred.

Response 8: The time horizon within which the consumption of fried foods has increased since many centuries ago. Please see the introduction of the revised manuscript on page 1, line 39-40.

Point 9: When talking about the microwave frying process, it is necessary to develop this new concept in more depth, as it is written there is no specification for the development of the process or the characteristics of the devices used for this purpose. The article would improve the interest of the readers if it improves this point.

Response 9: The explanatory texts of the characteristics of the microwave assisted frying device were added at the introduction of the revised manuscript page 1, lines 44-45, to specify the development of the process. Thanks for the great suggestion.

Point 10: They could specify the ranges of temperatures and pressures used in vacuum frying.

Response 10: The operating temperature range and pressure used in the vacuum frying process were added in the introduction on page 2, line 51 of the revised manuscript.

Point 11: When it is mentioned that there are few works regarding new technologies for frying pork rinds, it could include a reference to support that phrase.

Response 11: We have cited a reference related to puffed pork skin to support not many studies on the fried pork skin. Please see the introduction of the revised manuscript on page 2, line 54.

Point 12: In point 2.2. It is necessary to review the wording, the samples were fried at times xx, xx, and the samples were withdrawn from the frying process at the time intervals already mentioned?
Response 12: We have revised the sentence in point 2.2 as “Pork skin samples were fried at 0-5 min, and the samples were withdrawn from the frying process at the time intervals, 0.5, 1, 1.5, 2, 3, 4, and 5 min.” Please see the revised section 2.2 on page 2, lines 73-74. Thanks for the comment.

Point 13: It is not clear if the oil draining process was the same for all types of frying experiments?

Response 13: The pork rinds were centrifuged after traditional- and microwave assisted-frying under the atmospheric condition. The vacuum-fried samples were centrifuged under vacuum conditions (0.058 Mpa) to remove extra oil. Thanks for the comments. (Please see the Materials and Methods section 2.2 of the revised manuscript on page 2, lines 80-87).

Point 14: Regarding the microwave frying process, there is nothing clear about the device used, if this is an experimental setup or a commercial device, layout, quantity of oil used, etc? In addition, it is mentioned that the power 3 of the equipment was used, which is not precise, it would be convenient for them to be expressed as a percentage of power in relation to 100% of power or also what power they actually used.
Response 14: For microwave assisted frying, an industrial microwave fryer (MF-1K, Chin Ying Fa Mechanicalind Co., Ltd., Chang Hua Hsien, Taiwan) was used. Sixteen litters of refined palm olein oil were preheated at 180 °C and fried at an intensity setting of P3 (The frying chamber was subjected to 10 sec microwave on and 5 sec microwave off cycle during 5 min frying) The microwave power was set at 2500W. Thanks for the great comment. Please see the revised abstract on page 2, lines 76-79.

Point 15: It is not clear whether the pigskin pieces used for the experiments were of the same characteristics.

Response 15: The pork skins collected from same species and season used for the experiments were of similar characteristics. The phrase was added to section 2.1, the second sentence of Materials and Methods of the revised manuscript on page 2, lines 65-66. Thanks for the suggestion.

Point 16: In the moisture determination method 984.25 is mentioned but the year/version of the AOAC used is not established.

Response 16: We are sorry for the mistake. The moisture determination method 984.25 was established in 1984. The AOAC method was cited as reference #14 and revised at reference #14. Please see the reference section of the revised manuscript on page 15, lines 481-482. Thanks for the great comment.

Point 17: In the breaking force tests, it is not explained in sufficient detail how these measurements were developed.

Response 17: The breaking force of fried pork skins tested conditions was added to explain how this text was developed and for researchers who could repeat evaluating the texture of fried pork skin. Please see section 2.5 Materials and Methods of revised manuscript lines 128-131 on page 3. Thanks for the great suggestion.

Points 18: In all equations, the number corresponding to each one is shown twice.

Response 18: The number corresponding to each one is revised to once. Please see the Materials and Methods of the revised manuscript on page 3.

Point 19: It is necessary to specify in the color measurements which was the illuminant and the observer used.

Response 19: The definition and measurement of the illuminant CIEL*a*b* were added at section 2.7 of Materials and Methods. The values range and appearance of L*, a*, and b* were also introduced. Please see section 2.7 of the revised manuscript on page 3, lines 141-146.

Point 20: The description of the procedures for microscopic observations is incomplete.

Response 20: The detailed operating procedures for fried pork skin observations used in SEM were added at the Materials and Methods section on page 4, lines 157-163 of the revised manuscript. Thanks for the comment.

Point 21: It is necessary to review the scales especially of time, for example 1 A and B are the same but D is presented differently for the same time interval.

Response 21: We have added the optimal frying time of each sample on Figure 4 caption based on its moisture content (3-5g/100g, wb) is similar and suitable for eating in the revised manuscript. Please see the text highlighted in red on page 11 (Figure 4). Thanks for the suggestion.

Point 22: It is not explained anywhere why at 1 C there are no measurements for vacuum frying.
Response 2: Sentence “The oil absorption kinetic model of vacuum frying could not be predicted by Equation (1) due to the oil content of fried pork rinds did not change significantly after 4 min of frying (Figure 1A). Therefore, Figure 1C only has 2 predicted curves appear and not 3 like in the other graphs of Figure 1” was added to explain Figure 1C on page 6 (lines 211 to 214). Thanks for the great suggestion.

Point 23: Check if the equations used for the kinetic models have any proper name and if there are more equations of similar phenomena such as dehydration that can be applied.

Response 23: We have used another oil absorption kinetic model proposed by Moyano and Pedreschi (2006) and calculated the coefficient of determination (R2=0.97 for deep-fat frying) as foolowing figure, similar to that of (0.95) the kinetic model proposed by Krokida et al. (2006). Therefore, we only showed the equation of the kinetic model proposed by Krokida et al. (2006). Thanks for the suggestions, and we very much appreciate your consideration on this matter. (Please see the Results section at page 6 line 211-214 of the revised manuscript).

Moyano, P. C., & Pedreschi, F. (2006). Kinetics of oil uptake during frying of potato slices: Effect of pre-treatments. LWT-Food Science and Technology39, 285-291.

Fig.  Oil absorption model of fried pork rind with different frying methods (Moyano and Pedreschi, 2006). Experimental data and predicted by model Eq. (3).

NR: Nonlinear Regression

Eq. (3): ((Oe-O0)×K×t)/(1+K×t)+O0

Point 24: In figure 2 the concept of water activity must be written with a lowercase letter and use the subscript w.
Response 24: The water activity in Figure 2 was revised to “aw” with a lowercase letter and used the subscript w. Thanks for the great comment. Please see the revised Figure 2 on page 6.

Point 25: In table 1 there is no information for vacuum frying? Why the document must be uniform ...?

Response 25: The oil absorption kinetic model of vacuum frying could not be predicted by Equation (1) due to the oil content of fried pork rinds did not change significantly after 4 min of frying (Figure 1A). Therefore, Table 1 only has 2 predicted curves appear and not 3 like in the other graphs of Figure 1. Thanks for the great comment.

Point 26: In 3.2 they mention the contribution of frying to aroma and flavor, but didn't they really measure it?

Response 26: We have done a sensory evaluation of fried pork rind with different frying processing methods based on moisture content by 74 untrained panelists. The aroma, flavor, color, texture, greasy and overall acceptability of fried pork rinds were evaluated. However, the aroma and flavor are not significantly different. Therefore, we did not show the data and just mentioned sensory evaluation data not shown. Please see section 3.2 of the revised manuscript on page 7, line 235. Thanks for the great comment.

Point 27: What is the contribution of comparing with carrot chip, could you develop this point further and include other similar products in the discussion.

Response 27: The breaking force of carrot chips decreased with increasing frying temperature and vacuum degree (Fan et al., 2005). We add the sentence “And the breaking force of carrot chip could decrease by increasing the vacuum frying oil temperature.” We tried to explain the breaking force of fried pork skin could be decreased by increasing vacuum frying oil temperature or frying the pork skin to the moisture content close to 6% (wb). Please see section 3.2 of the revised manuscript on page 7 (lines 238-239). Thanks for the great comment.

Points 28: Check the figure number, figure 1B does not refer to the breaking force.

Response 28: The word “Figure 1B” was deleted in section 3.2 line 239 due to the figure was not referring to the breaking force. Please see the Results section of the revised manuscript on page 7, line 239.

Point 29: The discussion regarding the puffing ratio is very poor, the article would improve if it is broadened.

Response 29: The comparison in frying time of minute 4, in which the data appear for all the proposed operating conditions, was added at the discussion section on page 8, lines 249-252 of the revised manuscript. Thanks for the great comment.

Point 30: Table 3 mentions traditional frying? Traditional deep-frying experiments were done, if so it should be described in the methods part.

Response 30: We are sorry for sometimes using the phrase “traditional frying” and sometimes “traditional deep-frying.” Therefore, all the phrase “traditional frying” was revised to “traditional deep frying,” and the phrase “microwave frying” was revised to “microwave assisted frying” for all Tables of the revised manuscript. Thanks for the comment.

Point 31: It is important to review and consider that the color of this type of product is strongly influenced by the porosity, translucency, and brightness of the samples. The discussion of color needs to be improved.

Response 31: We have revised the last paragraph of section 4.2 related to the color of puffed pork skin to support our study on the fried pork skin. Hopefully, it has great improvement. Please see the Discussion section of the revised manuscript at lines 379-383 on page 13.

Point 32: What does table 4 refer to with cooked? Nowhere is there a talk of a cooking process?

Response 32: We have revised the word “cooked” to “Dried pork skin,” and the phrase “Traditional frying” to “Traditional deep frying,” “Microwave frying,” to “Microwave assisted frying.” Please see the marked red text of Table 4 of the revised manuscript on page 9.

Point 33: To make the color analysis and its results more attractive, you could use a free tool like Adobe Color Wheel to better represent the evolution of the color of the samples.

Response 33: We have added the supplementary Figure S1 Appearance of fried pork rind by different frying methods and frying time to show the evolution of the color of the sample. If it is not good enough, we will use a free tool such as Adobe Color Wheel to demonstrate the color of the fried sample change. Please see the supplementary Figure S1 of the revised manuscript on page 8, line 256 as follows:

Point 34: Review the relevance of table 5, it is important not to repeat data.

Response 34: We have added three extra sentences to discuss Table 5 as “Microwave-assisted frying method demonstrated it took less time to decrease the moisture content of fried pork skin to 3% and obtain a lighter color appearance (Tables 4 and 5). Although it was not the lowest oil content of fried pork. It significantly reduced oil content of fried pork skin, frying time, energy and produced high puffing ratio product”. Thanks for the suggestions, and we very much appreciate your consideration on this matter. (Please see the discussion section 4.2 of the revised manuscript on page 13, lines 379-383).

Point 35: The analysis of the microphotographs is very poor, it can be expanded and relate this to the texture / strength, oil content and moisture, in the same way the correlation analysis can be improved.

Response 35: Three sentences “Non-uniform and irregular surface structures of fried pork skins were observed on the traditional deep frying and microwave assisted frying (Figure 4A and 4C) comparing to the surface of vacuum fried pork skin (Figure 4E).

There is a high positive correlation between appearance, flavor, texture, and overall acceptability. And there is a high positive correlation between unpleasant smell and greasy sensation of fried pork rinds (Table S4; p < 0.01)” was added to sections 3.3 and 3.4 on pages 10 and 12.

Several discussion sentences were added in sections 4.3 and 4.4. “However, we found more cracks and holes were on the surface of traditional deep-fried pork skin (Figure 4B) compared to that of microwave assisted fried pork skin (Figure 4D). Microwave-assisted frying took less time to fry, and fewer cracks and holes on the surface of fried pork skin might explain it significantly absorbed less oil during frying. The SEM of vacuum fried pork skin demonstrated more regular and uniform hole structures (Figure 4E and 4F). We believed the absorption of frying oil, collagen gelatinization, dehydration, protein denaturation, and spontaneous water evaporation during low-temperature vacuum frying at 120°C and 0.058 Mpa created these structures. The frying temperature could go further higher under vacuum to shorten frying time.

It indicated the desirable quality properties such as appearance, flavor, and texture were associated with consumer perception. Therefore, it needs to understand to produce a fried pork skin with crispy texture and light appearance but low-fat content”. We hope this additional information will have significant improvement. Please see the Results sections 3.3 and 3.4 and Discussion sections 4.3 and 4.4 of the revised manuscript.

Point 36: In some parts of the speech of steam cooked denatured pork skin… which can be misleading, it is necessary to standardize all with pork rinds.
Response 36: We have revised the third paragraph in section 4.1 as “The evaporated steam water and hot oil cooked the denatured pork skin as it escaped through the pores because of internal pressure. When the fried pork skin was taken out of the hot oil, the temperature dropped and reduced steam production inside pork skin, and the fried pork skin temperature started to cool down. As the internal pressure decreased, the voids created by water evaporation through the pores absorbed the frying oil into the outer layer.” Please see the revised texts in section 4.1 on page 12, lines 309-312. Thanks for the suggestion.

Point 37: The conclusion can be improved by including the possibility of developing healthier foods, with less oil consumption and also the different use of energy and possible gaps to investigate such as energy aspects, stability and sensory evaluation of the products obtained.

Response 37: The last sentence of the conclusion was revised to “Microwave assisted-frying significantly decreased oil content and frying time of pork rinds to develop healthier foods and less energy spent and kept the best appearance and texture; thus demonstrating the most recommended technology for the fried pork rind industry.” Thanks for the great comments. (Please see the Conclusion section of the revised manuscript on page 14, lines 429-432).

The manuscript has been resubmitted to your Journal. We look forward to your positive response.

Sincerely yours,

Wen-Chieh Sung, Ph.D.

Professor

Department of Food Science

National Taiwan Ocean University

Reviewer 2 Report

The paper entitled "Kinetics of moisture loss and oil absorption of pork rinds during deep-fat, microwave and vacuum-frying" it´s very interesting and updated research because handled aspects like the different methods of cooking pork rinds. The title is simple but precise and the structure of the paper is in general correct and easy to read. Nevertheless, it is advisable to change some aspects like as:

  • In the abstract are reported various results in percentages of fat content (%), but is not mentioned the change of the variables with their respective units. For better comprehension of the results, it´s suggested to place these changes numerical with their respective units.
  • In the section "2.2 Traditional, microwave and vacuum-frying" is mentioned: “at an intensity setting of P3”. But P3 is not determined, it's suggested to specify previously in the text this parameter.
  • It is suggested to review the text in the English language.
  • It is recommended to justify in the text, why the operating conditions for vacuum fried were modified with respect to those previously proposed. This with the sole purpose of being able to compare the different methodologies mentioned.
  • It is recommended to mention the numbers of all equations in the text, before their appearance in the document. And do not repeat the numbering of the equations.
  • In equation 4 Color Change, it is suggested to define the parameters that here appear (L *, a * and b *).
  • The authors are recommended to mention, why in Figure 1C only 2 curves appear and not 3 like in the other graphs of this Figure 1.
  • The authors are suggested to mention in Tables 1 and 2, what the subscripts a, b and c represent, from a statistical point of view.
  • Specify why the results in table 1 were defined with variables different from those that appear in equation 1.
  • We recommend that the authors make a comparison of the results of Table 3, with the data reported in minute 4 because it is in this minute that the data appear for all the proposed operating conditions.
  • We recommend more profundity in the discussion of the results and the conclusions.

Author Response

Responses to Comments and Suggestions for Authors

Foods

Title: Kinetics of moisture loss and oil absorption of pork rinds during deep-fat, microwave assisted and vacuum-frying

Dear Reviewer #2

Thank you for your instruction on revising abstract, materials and methods, results and discussion. conclusion. We have rewritten the manuscript by Dr. Lin accordingly, and replied to comments and suggestions for authors are listed below point:

Reviewer #2’s comments and suggestions:

Point 1: In the abstract are reported various results in percentages of fat content (%), but is not mentioned the change of the variables with their respective units. For better comprehension of the results, it´s suggested to place these changes numerical with their respective units.

Response 1: We have revised the abstract and conclusion of the revised manuscript for the fat content unit of fried pork rind as g/100g dry weight basis (db). Please see the red marked text of the abstract and conclusion of the revised manuscript at pages 1 and 14.

Point 2: In the section "2.2 Traditional, microwave and vacuum-frying" is mentioned: “at an intensity setting of P3”. But P3 is not determined, it's suggested to specify previously in the text this parameter.

Response 2: For microwave assisted frying, an industrial microwave fryer (MF-1K, Chin Ying Fa Mechanicalind Co., Ltd., Chang Hua Hsien, Taiwan) was used. Sixteen litters of refined palm olein oil were preheated at 180 °C and fried at an intensity setting of P3 (The frying chamber was subjected to 10 sec microwave on and 5 sec microwave off cycle during 5 min frying) The microwave power was set at 2500W. Thanks for the great comment. Please see the revised abstract on page 2, lines 76-80. Hopefully, it has great improvement.

Point 3: It is suggested to review the text in the English language.

Response 3: We have rewritten and rechecked the manuscript in our article carefully as red marked texts in the revised manuscript. Although, we have asked Enago, an editing brand of Crimson Interactive Inc., for English language, grammar and spelling as attached certificate of editing. We asked Dr. Lin to rechecked the revised manuscript again. If it is not good enough, we will find another professional English editing company to help us. Thanks for the suggestions and we very much appreciate your consideration on this matter. (Please see the revised manuscript).

Point 4: It is recommended to justify in the text, why the operating conditions for vacuum fried were modified with respect to those previously proposed. This with the sole purpose of being able to compare the different methodologies mentioned.

Response 4: The vacuum fryer conducted for vacuum-frying operating conditions was at 120°C and 0.058 Mpa for our previous frying fish skin research. Therefore, the frying temperature and vacuum conditions used for this study for this fryer. So, its frying temperature is different to the other two methods. Vacuum-frying usually works at reduced temperature (100-144°C).

Point 5: It is recommended to mention the numbers of all equations in the text, before their appearance in the document. And do not repeat the numbering of the equations.

Response 5: The number of equation was mentioned behind the kinetic model equation such as “The oil absorption kinetic model of vacuum frying could not be predicted by Equation (1) due to the oil content of fried pork rinds did not change significantly after 4 min of frying (Figure 1A).” Please see page 6 lines 211-214 and page 12 lines 315 and 329 of the revised manuscript.

Point 6: In equation 4 Color Change, it is suggested to define the parameters that here appear (L *, a * and b *).

Response 6: The definition of CIEL*a*b* was added at section 2.7 of Materials and Methods. The values range and appearance of L*, a* and b* were also introduced. Please see the section 2.7 of the revised manuscript at page 3 lines 141-146.

Point 7: The authors are recommended to mention, why in Figure 1C only 2 curves appear and not 3 like in the other graphs of this Figure 1.

Response 7: Sentence “The oil absorption kinetic model of vacuum frying could not be predicted by Equation (1) due to the oil content of fried pork rinds did not change significantly after 4 min of frying (Figure 1A). Therefore, Figure 1C only has 2 predicted curves appear and not 3 like in the other graphs of Figure 1” was added to explain Figure 1C at lines from 211 to 214 at page 6. Thanks for the great suggestion.

Points 8: The authors are suggested to mention in Tables 1 and 2, what the subscripts a, b and c represent, from a statistical point of view.

Response 8: The subscript letters of values for statistical different explanatory text were added to the footnote of Tables 1 & 2. Please see the revised Tables 1 & 2 at page 7 of the revised manuscript.

Point 9: Specify why the results in table 1 were defined with variables different from those that appear in equation 1.

Response 9: The explanatory texts of results in Table 1 were added at page 6 lines 218-221 and page 12 lines 319-324 to discuss variables different from those that appear in equation 1. Thanks for the great comments.

Point 10: We recommend that the authors make a comparison of the results of Table 3, with the data reported in minute 4 because it is in this minute that the data appear for all the proposed operating conditions.

Response 10: The comparison in frying time of minute 4 was added at the discussion section at page 8 lines 249-252 of the revised manuscript.

Point 11: We recommend more profundity in the discussion of the results and the conclusions.

Response 11: We have revised the Results section and added more discussion and added several main thoughts from our results to conclusion. Please see the revised Results, Discussion and Conclusion sections of the revised manuscript.

The manuscript has been resubmitted to your Journal. We look forward to your positive response.

Sincerely yours,

Wen-Chieh Sung, Ph.D.

Professor

Department of Food Science

National Taiwan Ocean University

Round 2

Reviewer 1 Report

The authors of the article gave answers to all comments.

Reviewer 2 Report

The document shows improvement over the previous version. Therefore, I suggest its presentation be included in this journal.